# Effectiveness of an Innovative Sensory Approach to Improve Children’s Nutritional Choices

**DOI:** 10.3390/ijerph18126462

**Published:** 2021-06-15

**Authors:** Domenico Meleleo, Giovanna Susca, Valentina Andrulli Buccheri, Giovanna Lamanna, Liliana Cassano, Valeria De Chirico, Sergio Mustica, Margherita Caroli, Nicola Bartolomeo

**Affiliations:** 1Social Promotion Association Lifestyle Studium, 76012 Camosa di Puglia, Italy; domenico.meleleo@gmail.com (D.M.); dott.ssagiovannasusca@gmail.com (G.S.); valentina.ab@gmail.com (V.A.B.); gio.lamanna81@gmail.com (G.L.); cassanoliliana@gmail.com (L.C.); valeriadechirico@gmail.com (V.D.C.); 2Italian Society of Obesity, 56125 Pisa, Italy; margheritacaroli53@gmail.com; 3Laboratorio delle Idee s.r.l., 60044 Fabriano, Italy; sergio.mustica@gmail.com; 4Department of Biomedical Sciences and Human Oncology, University of Bari Aldo Moro, 70124 Bari, Italy

**Keywords:** prevention, eating habits, nutrition education, public health, nutrition-conscious knowledge

## Abstract

A case-control study was conducted to investigate the effectiveness of the Edueat^®^ Method, through experiential workshops focused on the use of all 5 senses. In two different primary schools in the same city, questionnaires were administered in two months with a follow-up one year later. Participants: 119 children (age 8.2–9.0) chosen randomly; control group 66 (55.5%). Seven lessons of 2 h each were held in the schools by experts of the Edueat^®^ method and seven extra lessons by the teachers. The main outcome measures were the children’s changes in their approach and attitude towards their eating habits. The answers were grouped with factor analysis and summarized through scores. Repeated-measures analysis of variance was conducted in order to identify the relationships between scores and treatment over time. At the end of treatment, the intervention group showed a significant appreciation towards healthy foods (+4.15 vs. −0.05, *p* = 0.02) and a greater capacity in identifying foods which are very good for the health (+15.6 vs. +14.4, *p* = 0.02). In conclusion, the Edueat^®^ method was found to be particularly promising in transmitting knowledge of those foods which are healthy. Greater involvement of teachers and parents is crucial.

## 1. Introduction

For the prevention of non-communicable diseases (NCD), it is necessary to implement food education programs in the pediatric age.

To prevent obesity and other NCD, in the age of development (0–18 years), education interventions to a healthy diet must be designed, implemented and monitored by a multidisciplinary team which should take into account medical, psychological and pedagogical scientific evidence [1,2]. These interventions, thus, have better chances of being effective if they extend their range of action beyond the didactic intervention on the pediatric age target, involving their most important reference figures (i.e., teachers, family, sports centres, cultural centres) [3,4]. Family environment and socioeconomic status influence both the present and future food choices of the child, as well as the effectiveness of nutritional education interventions [5,6,7,8].

The didactic phase must thus be based mainly on experiential methods and not only on theoretical lessons [3].

In literature, the best results for food education interventions report mainly the increase in fruit consumption and, to a lesser degree, that of vegetables. Whereas, in the reduction of BMI in obese subjects, very few data are found and these deal mainly with long-term intensive programs [6,9,10,11,12,13,14] especially if they also provide an educational intervention aimed at increasing both the quantity and quality of physical activity.

Studies relating to food education converge in affirming that the development of a preference for foods is associated with the frequency of exposure to them and, consequently, with the increased consumption of these [15]. Furthermore, the style adopted by the parents in accompanying the children in the discovery and formation of preferences and eating habits is crucial. There is evidence that the presence of a positive emotional climate during meals is associated with a greater preference for food, as well as the use of the senses to capture the attention of children on the foods to be consumed, for example, by composing a figure with the fruit [16,17,18]. Therefore, early exposure to fruit and vegetables is important for their consumption to be frequent. Approaching food through the senses and in a serene relational climate is a powerful tool for learning these good eating habits.

It is important to establish the age at which it would be most effective to intervene. The first years of life are decisive for learning taste and eating habits. In this study was experimented an intervention with a pedagogically in-depth method with a sociocognitive approach, as per national guidelines and international studies [2,19,20]. Therefore, the Edueat^®^ method was used, which comes with a text designed for 8–9-year-olds. Furthermore, the project was of an educational type, designed for schools, so we opted for older children in the middle of elementary school. In this way, there was the guarantee that the participants had already created a meaningful relationship with the teachers, who are an indispensable reference and guide for the assimilation of the contents of the project. The Edueat^®^ method aims to provide families with a practical tool to help them in this challenging task. The method includes fun games to motivate children to use their senses in the exploration of food and involving parents in this journey of discovery.

The aim of this case-control study was to test the effectiveness of an innovative pedagogical approach, the Edueat^®^ method, to improve children’s food choices. The Edueat^®^ method uses workshops based on the involvement of all five senses to make children acquire a better and more aware knowledge of food.

## 2. Materials and Methods

The study was carried out between November 2016 and November 2017. Informed consent was obtained from all parents of the participants in this project. The study was carried out in accordance with the Helsinki Declaration guidelines and was approved by the Parents–Teachers Association of the schools involved and by the teachers directly involved in the study. The Edueat^®^ method was devised and developed by the Laboratorio delle Idee-Fabriano (AN), Italy, with the collaboration of the Department of Educational Sciences of the University of Macerata, Italy, as a tool for the promotion of food education both in the school and in the family [21,22]. Through games and the use of the 5 senses, this method aims at promoting the acquisition of the child’s own identity, the increase of social relationships and culture, for children in the pediatric age and their families. In fact, with the acquisition of identity, the choices are more aware and less influenced from the outside. The main tools of the method are two volumes, one to be used by the children and one to be used by the adults (parents and teachers). The first volume contains songs, poems in rhyme, fairy tales, interactive readings, games and exercises designed with a language suitable for children aged 8–9. The second volume contains indications for good practices: explanations of concepts regarding health, the use of the 5 senses, as well as instructions for games and procedures useful in reinforcing the effectiveness of the method in the family environment. These indications are both tools for food education, as well as tools to improve the parent–child relationship. The Edueat^®^ method guides participants through play and engages all of the five senses (taste, smell, sight, hearing and touch) using the “GOVUT” characters: GUSTINO, the elf, embodies the sense of taste; OLFAT, the dwarf with the big nose, embodies smell; VISTELLA, the blue fairy, embodies the sense of sight; UDINO, the pointy-eared leprechaun, embodies the sense of hearing and, finally, TATTONE, a small ogre with large hands embodies the sense of touch. Moreover, the “vocacibario”, a dictionary containing only adjectives related to food, was used.

### 2.1. Population

The project was carried out involving the participants and their families of some of the classes of the “Mauro Carella” primary school of Canosa di Puglia (BT), Italy. The survey involved 119 respondents (aged 8.2–9.0 years old) attending the school; of these, three classes, comprising a total of 66 participants (55.5%), followed the Edueat^®^ method. Three other classes, matched by grade and taken from a randomly selected school among the others located in another school district of the same city, were used as a control group. This division of location in different school districts was necessary to reduce possible bias due to the mutual influence between the two groups. The difference in sex and age between cases and controls was not statistically significant (Table 1). The classes in the control group were given only a broad outline of the project, in order to obtain informed consent from the parents to the participation and engagement of their children.

### 2.2. Questionnaires

Three questionnaires were chosen to assess the effectiveness of the intervention. The questionnaires were developed by the Public Health Department of the Emilia Romagna region with the aim of carrying out a nutritional surveillance action in elementary schools (for 9-year-olds) of the Cesena Medical Health Services (2007). For these questionnaires, there are preliminary validation tests (S2, p. 11). The questionnaires included various topics, such as participants’ preferences and eating habits and their attitudes towards food. The text of the questionnaires can be found in Appendix A. Questionnaires were administered in class by reading the question together and explaining it. The participants marked the answer with a pen. Anonymity was guaranteed through a system which assigned each child an identification number known only to the investigators. Questionnaire A consists of 10 questions with true/false answers, to investigate the children’s eating habits and attitude towards food. Questionnaire B investigates food preferences; 32 healthy foods are listed, and for each food, the participants had to choose from three possible answers: “I like it very much”, “I like it so-so” or “I do not like it at all”. To avoid complacent responses from the participants, the questionnaire did not specify that all the foods listed were healthy. Questionnaire C verifies the ability of the respondents to recognize healthy and unhealthy foods; 32 foods and drinks are listed, and for each of these foods, the participant had to choose from three possible answers: “very good for your health”, “good for your health” and “not good for your health”.

### 2.3. Procedure

The study consisted of the following phases:(1)On 11 November 2016, a preliminary meeting was held with the teachers and the parents of the children of the intervention group, to present the Edueat^®^ method and the phases of the project. During the meeting, the author of the book illustrated the contents, the pedagogical assumptions of the method and provided didactic material to stimulate the use of the senses during teaching. All the six teachers in the intervention group were present as well as 38 parents.(2)Time 0 (T0). On 15 November 2016, questionnaires were administered both in the intervention group and the control group, to assess the children’s preferences and eating habits as well as their attitude towards food. The questionnaires were administered by qualified experts of the Edueat^®^ method with the help of the teachers.(3)Expert intervention. In the period between 15 November and 24 November 2016, seven lessons were carried out for each class in the intervention group, each lasting two hours, including a theoretical part and a ludic part. These lessons were held both in the classroom and during meals at the school canteen. The interventions were carried out by the Edueat^®^ method experts with the help of the teachers. A brief description of the methods of carrying out the theoretical and practical lessons is reported in Appendix A.(4)Time 1 (T1). On 1 December 2016, post-intervention questionnaires were administered to assess the effectiveness of the intervention. The questionnaires were again administered by qualified experts of the Edueat^®^ method with the help of the teachers, both in the intervention group and in the control group, one week after the end of the lessons held by the experts.(5)Teacher intervention. In the period between December 2016 and May 2017, teachers held seven extra lessons in the intervention classes. The teachers had been previously instructed in the use of the Edueat^®^ method by the Edueat experts.(6)Post-intervention meeting. On 20 May 2017, a meeting with the Edueat^®^ method experts and with the teachers and parents of the participants of the intervention group was held in order to discuss the experience just ended. They were invited to create formal and informal opportunities to stimulate the children to use the senses in discerning between healthy and unhealthy foods. Six teachers and only 14 parents were present.(7)Time 2 (T2). On 20 May 2017, to assess the effectiveness of the intervention, the same questionnaires were administered at the end of the school year. The questionnaires were administered both in the intervention group and in the control group by qualified experts of the Edueat^®^ method with the help of the teachers.(8)Time 3 (T3). In November 2017, one year after the intervention of the experts, the same follow-up questionnaires were administered to assess the effectiveness of the intervention. The questionnaires were administered by qualified experts of the Edueat^®^ method with the help of teachers, in both the intervention group and in the control group.

### 2.4. Statistical Analysis

Descriptive statistics focused on summaries of response frequencies and response consistency shown as number and percentages. Exploratory factor analysis was applied to questionnaires A and C to examine the relationships between survey items and the underlying structure of the survey. If the answers of the participants on different subsets of questions are more closely related to each other, then the number of variables can be reduced to a single variable or factor which is a combination of the same [23,24]. For questionnaires B and C, we assigned a score from 0 to 2 for each item and calculated a total score. The highly incomplete questionnaires (percentage of missing answers above 25%) were removed from the analysis. Although analyzing only complete cases has the advantage of simplicity, the information contained in the incomplete cases is lost. This approach also ignores possible systematic differences between the complete surveys and the incomplete surveys. Therefore, in the slightly incomplete questionnaires (percentage of missing answers between 0% and 25%), we performed a “multiple imputation” procedure for missing data and then calculated the total score.

In the second stage of analysis, repeated-measures analysis of variance was used: for questionnaire A, to identify relationships between the estimated factors and the treatment by time; for questionnaires B and C, to identify relationships between the total scores and the treatment by time. Multiple comparisons between estimate scores were adjusted according to Tukey. All tests of statistical significance were two-tailed, and *p*-values of less than 0.05 were considered statistically significant. Statistical analysis was performed by SAS software (SAS Institute, Cary, NC, USA).

## 3. Results

### 3.1. Questionnaire A. Child’s Eating Habits and Attitudes towards Food

The average percentage of missing answers to each question was 7%. Only questionnaires with a response rate greater than 75% were selected (*n* = 58 for cases, *n* = 44 for controls), and for these the missing answers were assigned with a Multiple Imputation process. Through factorial analysis, the 10 questions of Questionnaire A were reduced to two factors. The first factor corresponds to the sequence of the following six answers, A2-true, A4-false, A5-true, A7-true, A9-false, A10-true, and was summarized with a score ranging from 6 to 12, assigning 2 points to answers that matched the sequence, 1 point otherwise. This factor identifies the difficulty in making participants eat and, above all, in varying their diet, with the score increasing as the difficulty decreases. Similarly, the second factor corresponding to the sequence of four answers, A1-true, A3-true, A6-true, A8-true, was associated with a score ranging from 4 to 8. This second factor characterizes the tendency to eat between meals, focusing more on quantity than quality, and the score increases as this propensity decreases.

The results of the repeated-measures analysis of variance applied to the two factors are shown in Table 2.

Paradoxically, the first factor shows, at T1, an increase in the score and therefore a reducing difficulty in making children eat and vary their diet in the control group, while the score is substantially unchanged in the intervention group; however, the effect of the treatment is not significant (−0.04 in treated vs. +0.52 in untreated, *p* = 0.085). At T2, compared to T0, the score increases in both the treated and the untreated, and the difference is obviously not significant (+0.24 in treated vs. +0.58 in untreated, *p* = 0.317).

The second factor (tendency to eat outside of mealtime, focusing more on quantity than quality) shows that the intervention has no significant impact on the responses at T1 and T2 compared to the initial T0. The tendency to eat outside of mealtime decreases (the score increases) in both groups at T1 (+0.22 in treated vs. +0.12 in untreated, *p* = 0.699) and then returns to the same score at T2 at the same levels as T0.

For both factor 1 and factor 2, there are no relevant effects at time T3 (data not shown).

### 3.2. Questionnaire B. Food Preferences

At T0, the average percentage of respondents was 93.3%; the highest rate of respondents was for questions B18 and B24 (94.5%), whereas the lowest was for question B01 (88.6%). At T1, the average percentage of respondents fell to 92.0% with a range of 92.8% (Questions B06, B12, B17, B24, B26) and B01 (84.8%). At T2, the average percentage rose to 94.3% (94.5% B06, 89.9% B01). At T3, the average percentage of respondents was only 85.6% with a minimum for question B01 (81.5%). For the subjects who had, for each time, a percentage of answers of at least 75% (24 out of 32 questions), the missing responses were imputed through a Multiple Imputation process. Then, the score was calculated for 97 respondents (56 cases and 41 controls) at times T0, T1 and T2, and for 87 respondents (51 cases and 36 controls) at time T3.

The foods in questionnaire B, which were used to assess the children’s appreciation of foods, represent all healthy food choices. A single score was created as the sum of the scores attributed to the 32 answers (2 points “very”, 1 point “so-so”, 0 points “not at all”) for values which can be considered directly proportional to the wholesomeness of the food preferences.

The effect of the intervention over time was estimated also on the basis of the class attended, using as adjustment variables, sex as well as age through the application of a Generalized Linear Model.

At each time, the average score of the subjects treated is not significantly different from that of the untreated (Table 3).

On the other hand, the general effect of treatment over time is statistically significant (*p* = 0.04). In particular, at T1 compared to T0, the average score increases more in the intervention group than in the control group, but the difference is not significant (+2.4 vs. +1.8, *p* = 0.72), while the effect of the treatment is statistically significant in the T0–T2 period with the score increasing in the treated while decreasing in the controls (+4.15 vs. −0.05, *p* = 0.02).

It is important to note that at T3, (one-year after the treatment) the average scores of the two groups (treated and untreated) are similar, 43.6 vs. 44.0 with the effect that the treatment is no longer significant with respect to T0 (*p* = 0.87).

### 3.3. Questionnaire C. Opinion on the Solubrity of the Different Foods

At T0, the average percentage of answers was 93.7%, with the lowest percentage of respondents to question C12 (91.1%). At T1, the average percentage of respondents fell to 92.5% (90.7% for question C32) and then rose to 94.5% at T2 (lowest frequency, 92.4% for C13). At T3, the average percentage of respondents was only 85.7% with a minimal number of responses for question C12 (83.2%). For the subjects who had, for each time, a percentage of answers of at least 75% (24 out of 32 questions), the missing responses were assigned through a Multiple Imputation process. Then, the score was calculated for 100 respondents (57 cases and 43 controls) at times T0, T1 and T2, and for 87 respondents (51 cases and 36 controls) after a year (T3).

The foods listed in Questionnaire C have contrasting health effects; therefore, no single score was created, but a factorial analysis was chosen based on the responses provided by the participants to subdivide foods into homogeneous groups. The factorial analysis has therefore allowed us to separate 32 items into 3 factors or groups. The first group contains the foods evidently considered “unhealthy” by the respondents: sugary drinks, cookies, sweets, chocolate, ice cream, hamburgers, snacks, potato chips, salami/mortadella, fruit juice, cakes and wine. The second group contains the foods that participants perceived as “good for health”: butter, meat, breakfast cereals, cheese, milk/yogurt, bread, pasta/rice, pizza, cooked ham, raw ham and eggs. The third group identifies foods considered extremely “healthy”: water, fruit, legumes, vegetable soup, olive oil, potatoes, fish, raw vegetables and cooked vegetables.

Score “0” was assigned to the response “not good for your health”, “1” to the response “good for your health” and “2” to the response “very good for your health”. Tree scores were calculated, one for each food group. A low score corresponds to a global awareness of the “harmfulness” of certain foods, while, on the contrary, a higher score is associated with the awareness of the benefits that certain foods have on health. Thus, a low score is expected for the first food group, while a higher score is expected for the other two groups.

In the first food group (factor 1), “*foods that are not good for your health*”, there is a greater awareness of food in the treated group compared to the control group at T1 compared to T0. In fact, in the intervention group, the score decreases by 1.3, while in the controls it drops by 0.01 but this difference is not statistically significant (*p* = 0.08); at T2, compared to T0, no effect of the treatment is observed as there is a decrease in the score of the untreated group as well (−1.2 intervention group vs. −1.5 control group, *p* = 0.69). In the exploratory analysis, we noted that, at T3, the scores of the intervention group (5.1) are higher than the control group (4.5) (Figure 1a).

For the second food group (factor 2), “*foods good for your health*”, the treatment has no significant effect. Between T0 and T1, the overall score of the 11 items fell from 13.4 to 13.0 in the treated group and from 13.2 to 12.8 in the control group (*p* = 0.95). At T2, it further dropped to 12.4 in the treated group and to 12.1 in the controls, with no significant effect with respect to T0 due to treatment (*p* = 0.94). The score rises slightly in both groups in the subjects assessed at T3 (Figure 1b).

The score of the third food group (factor 3), “*foods that are very good for your health*”, is expected to increase in the intervention group. However, the analysis shows no significant effect of treatment between T0 and T1 (*p* = 0.18). In fact, at T1, the score in the intervention group is almost the same as that of the controls (15.0). At T2, the score rises to 15.6 in the treated group, whereas it drops to 14.4 in the untreated group, thus highlighting a statistically significant effect of the treatment (*p* = 0.02) compared to T1. As with the other factors, the average scores between the two groups, again, at T3, tend to become closer (Figure 1c).

## 4. Discussion

The participants’ awareness of unhealthy foods does not change despite the educational intervention (questionnaire C); just as there is no change in eating behaviors (questionnaire A). These results are in line with most of the data found in literature. Studies show that, as a result of food education programs, there is mainly an increase in fruit consumption, but poor results on the other targets [6,9,10,11,12]. In addition, a constant involvement of families in educational programs is an essential factor in influencing the children’s attitudes and eating habits. This factor is lacking in this study because only two meetings were held with the parents and only a few of the parents actually attended these meetings. Moreover, the family environment and socioeconomic status of the children [3,5,6,7] were not investigated before these educational interventions.

The positive effect of intervention on the healthy food choices in questionnaire B and the increased knowledge of very healthy foods (third group in questionnaire C) are in line with what is usually reported in literature. The positive effect is obtained in function of the duration of the intervention and especially of the number of meetings held [2,4]. Most likely, an important positive effect was played by having meetings held by the teachers after those held by the experts. This choice increased the number of meetings with the children, and probably the teachers, with their authority and particular relationship with pupils, have the possibility of creating continuous and detailed nutritional education interventions for more effective learning. This, together with the loss of the effect of the treatment at T3, treatment one year after the end of the educational project, is in line with what has already been reported in literature and underlines the importance of planning continual education interventions for children, which should be long term and should actively involve teachers and families.

Through the application of the factorial analysis to the responses of sections A and C of the questionnaire, it was possible to identify the common traits in food choices and preferences reported by participants. The occurrence of these common traits, with regard to questionnaire C, correspond to what could have been a priori a distinction of the 32 foods in relation to their effect on health. We have thus found that the participants of the intervention and control groups are equally aware of what healthy foods are.

The application of the factor analysis to the results of questionnaire A, on the other hand, allowed two core features to emerge on the attitudes of the children towards food: one on the restrictiveness and selectivity of food and the other on impulsivity and unplanned consumption, for example by eating between meals. Both of these aspects represent early forms of food attitudes that in a more marked form can give rise to real eating behavioral disorders. In fact, both The Avoidant/Restrictive Food Intake Disorder and Anorexia Nervosa show particularly severe forms of food selectivity and restriction, just as Bulimia Nervosa and the Binge Eating Disorder are based on out-of-control food consumption characterized by binge eating [25]. However, the factors obtained through the application of factor analysis to questionnaires A and C should be validated in order to give them a more robust meaning, even on a statistical level.

The family is the first educational agency for each individual and has the task of showing and teaching children behavior patterns and conduct, appropriate in the different contexts of life. Nutrition, beside other behaviors and habits, is taught in the family ambient. There is ample evidence that already in early childhood the child develops preferences towards foods, attitudes and peculiar behavior, both healthy and balanced, as well as openly pathological or unregulated, in relation to the type of education and the relational dynamics that parents have established with them [8,16,26,27,28,29,30,31].

Several studies have shown that parenting feeding practices aimed at restricting children’s access to fatty foods or pressing them to consume fruit and vegetables are associated with the increased consumption of these unhealthy foods and the increased risk of childhood obesity [26,32]. Furthermore, this parental influence also includes the transmission of disturbed eating behavior from mothers to children [8]. On the other hand, virtuous parenting practices linked to the greater consumption of fruit and vegetables and less consumption of fatty foods have positive effects on children’s eating habits. These practices consist of creating an atmosphere in which children can choose foods, participate in meal preparation and eat them together in a pleasant atmosphere, encouraging them to regulate their food intake according to their sense of satiety, setting a good example as models to inspire them, constantly exposing them to a variety of healthy foods and explaining to them why it is important to eat them [26,33].

One possibility for these poor results obtained from the treatment in this study is due to the fact that the program was predominantly school-centered and there was little or no cooperation from the families. Food education and food choices are delicate and complex spheres in family dynamics where the family plays a key role in both the etiology and in the success of any educational and therapeutic intervention [4,34,35,36,37]. There is also the possibility that the intervention was not effective due to the not-very-early age of the participants.

## 5. Conclusions

A food education intervention lasting almost six months, in randomly chosen classes of primary school children, through the creation of educational workshops using five senses, in order to stimulate a more informed knowledge of food, has proven effective for raising awareness of what are wholesome, healthy foods. This positive effect at T2 (after six months of intervention) was based on the duration of the intervention and the number of workshops carried out in each class but then diminished at the follow-up, one year after the end of the intervention.

Treatment was found to be uninfluential in changing children’s eating attitudes and in their knowledge in recognizing unhealthy food. In addition, two distinct core features emerged towards food attitudes: one of restrictiveness and selectivity, and the other of out-of-control food consumption. Both of these aspects represent early forms of food attitudes that in a more marked form can give rise to real eating behavioral disorders. These results are in line with most of the data in literature and they emphasize, also for this type of educational intervention based on sensory workshop methods, the importance of the duration of the intervention, the number of workshops and that the family should be strongly involved in school nutrition education projects. Even earlier educational intervention could provide better results on eating habits.

## Figures and Tables

**Figure 1 ijerph-18-06462-f001:**
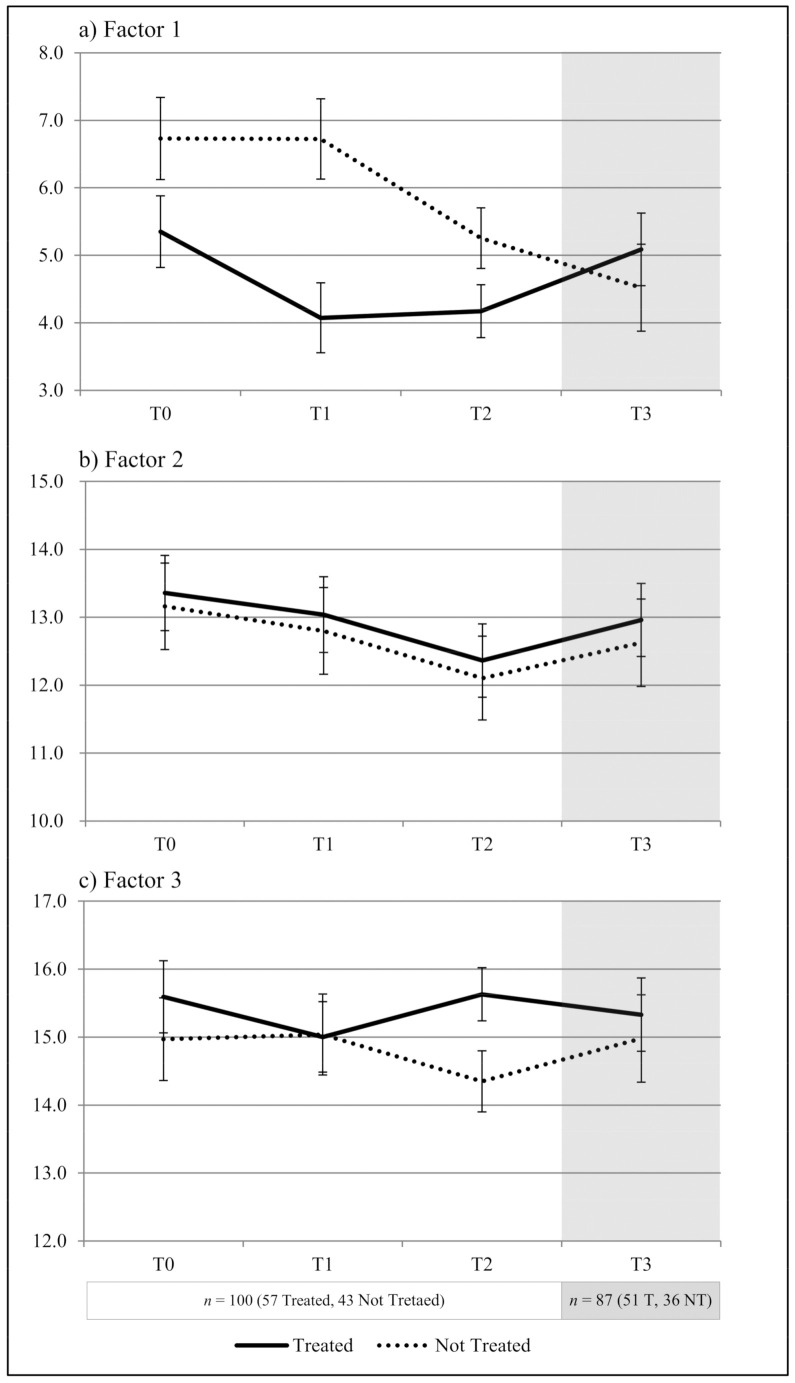
Least square means by time and treatment of scores for factors of questionnaire C. (**a**) Factor 1 “*foods that are not good for your health*”; (**b**) Factor 2 “*foods good for your health*”; (**c**) Factor 3 “*foods that are very good for your health*”.

**Table 1 ijerph-18-06462-t001:** Demographic characteristics of participants by intervention groups.

Group	Sex	*n* (%)	Age		
			Mean ± St.Dev.	Min	Max
Intervention	M	29 (43.9%)	8.18 ± 0.37	7.395	8.928
	F	37 (56.1%)	8.089 ± 0.369	7.124	8.838
Control	M	22 (41.5%)	8.23 ± 0.268	7.773	8.871
	F	31 (58.5%)	8.233 ± 0.357	7.480	8.827

**Table 2 ijerph-18-06462-t002:** Questionnaire A. Effect of nutrition education on eating habits and attitude towards food by participants.

Factor	Time	Intervention Group (Mean Score ± St.Err.)	Control Group (Mean Score ± St.Err.)	*p*
1 ^1^	T0	9.99 ± 0.21	10.08 ± 0.24	0.779
	T1	9.95 ± 0.23	10.6 ± 0.26	0.064
	T2	10.23 ± 0.22	10.66 ± 0.25	0.199
	Diff. T1 vs. T0	−0.04 ± 0.21	0.52 ± 0.22	0.085 *
	Diff. T2 vs. T0	0.24 ± 0.23	0.58 ± 0.23	0.317 *
	Diff. T2 vs. T1	0.28 ± 0.20	0.06 ± 0.20	0.421 *
2 ^1^	T0	5.81 ± 0.15	6.51 ± 0.18	0.003
	T1	6.03 ± 0.15	6.63 ± 0.18	0.011
	T2	5.82 ± 0.15	6.50 ± 0.17	0.002
	Diff. T1 vs. T0	0.22 ± 0.17	0.12 ± 0.17	0.699 *
	Diff. T2 vs. T0	0.01 ± 0.14	−0.01 ± 0.16	0.937 *
	Diff. T2 vs. T1	−0.21 ± 0.18	−0.13 ± 0.14	0.709 *

^1^ 58 Treated and 44 not Treated. Factor 1. Difficulty in making children eat and, above all, in varying their diet. (Range score 6–12). Factor 2. Tendency to eat between meals, focusing more on quantity than quality. (Range score 4–8). * *p-*values adjusted by Tukey for multiple comparison. T0: pre-intervention observation (baseline). T1: observation after expert intervention (one month after baseline). T2: observation after teacher intervention (six months after baseline). T3: observation one year after the first intervention of the experts.

**Table 3 ijerph-18-06462-t003:** Questionnaire B. Effect of nutrition education on food preferences by participants.

Time	Intervention Group (Mean Score ± St.Err.)	Control Group (Mean Score ± St.Err.)	*p*
T0 ^1^	40.18 ± 1.70	41.40 ± 2.00	0.639
T1 ^1^	42.62 ± 1.82	43.24 ± 2.13	0.824
T2 ^1^	44.33 ± 1.83	41.36 ± 2.14	0.291
Diff. T1 vs. T0 ^1^	2.44 ± 1.19	1.84 ± 0.89	0.721 *
Diff. T2 vs. T0 ^1^	4.15 ± 1.10	−0.05 ± 1.34	0.020 *
Diff. T2 vs. T1 ^1^	1.71 ± 1.26	−1.89 ± 1.28	0.060 *
T3 ^2^	43.61 ± 1.99	43.98 ± 2.43	0.907
Diff. T3 vs. T0 ^2^	3.43 ± 1.15	2.58 ± 1.59	0.874 *

^1^ 56 Treated and 41 not Treated. ^2^ 51 Treated and 36 not Treated. * *p-*values adjusted by Tukey for multiple comparison. T0: pre-intervention observation (baseline). T1: observation after expert intervention (one month after baseline). T2: observation after teacher intervention (six months after baseline). T3: observation one year after the first intervention of the experts.

## Data Availability

All data can be made available upon request.

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
