# Peer review of "Effectiveness of an Innovative Sensory Approach to Improve Children’s Nutritional Choices"

_ijerph, 2021, doi:10.3390/ijerph18126462_

Round 1

Reviewer 1 Report

This manuscript describes a sensory education program, Edueat, and discusses whether children who participated in this intervention showed nutritional knowledge and attitudinal changes toward healthy foods as compared to children who did not.  The use of a comparison group is a study strength as is the follow-up measurements.  I hope the following comments will be useful to the author(s).

Lines 28 -50 The introduction is fairly brief and did not seem well linked to the intervention.  I also thought the mention of NCD could just be eliminated. I would like to see more detailed evidence as to why the sensory approach would be effective to improving healthy eating by children. 

Lines 61 – 63.  Explain why the acquisition of identify is related to healthy eating.

Lines 64-67 – Please explain why this age group (8-9 year olds) are the intervention’s focus?  Noting that intervening earlier may be more effective would fit well in the introduction.

Lines 80-88  How were the other schools/school districts chosen?  Are they similar to the Mauro Carella primary school?  A table with the demographics of each group would be helpful.

Line 103 – please define “ludio”

Lines 113-11 9 What did the training of the teachers look like? 

Lines 120-123 – What is meant by strengthening the message and how was this done?  What changes occurred on the part of the teacher in the classroom?

Lines 130 – 145.  Where did the questions come from – where they based on a review of the literature or of other measures?  Have reliability and/or validity been established?  How were the foods chosen?  Who completed the questionnaires – the parent or the child?  Where were they administered – in the classroom?  How were the administered – orally or pen and pencil? 

Line 155-157 – Please provide a citation to support this method.

Lines 174-176 – Delete

Lines 178-184 – The results section was a bit hard to follow.  Perhaps put all the discussion of missing data together, and then all of the factor analysis together, and then end with the evidence around program impact.  I wasn’t sure why it was important to know what percentage answered each question – how is this related to effectiveness? 

Line 223 – remove the word “very” from “not very significant”

Did you consider testing for initial group differences (as the table with Factor 1 would suggest) and then controlling for these when you compared the outcomes?

Some discussion of the study limitations is needed, for example, lack of behavioral measures, the use of non-validated measures, the lack of measurement related to the 5 senses; no identification or statistical adjustments for possible group differences.

Lines 317-329 The authors seem to conclude that lack of family involvement is the reason for failure but there is no evidence to support that as this program was not compared to a home-based program.

Do the authors think that the 5 senses strategy needs to be changed in light of these findings? 

Comments on Supplemental materials:

Some more explanation of the activity sequences is needed. Does this describe one workshop?  An example of the nursery rhyme would be helpful.

I did not see questionnaire A in the supplemental materials.

Reviewer 2 Report

Dear Authors,

This manuscript covers a very interesting topic of educational intervention in elementary schools children, regarding healthy nutrition. Since the report on effectiveness of the experiment is important to be seen among other professionals from the field of educational and food intervention, I find the work significant and thus did a detailed analysis of what has been written, to support You in better publishing of these results.

However, the manuscript requires some significant changes in the sense of language style, grammar, punctuation and the meaning of sentences and also regarding the presentation and interpretation of the results.  The results from different section should be presented in the tables and with much more details and related to Supplementary material, which does not really respond to what has been written in the text. In supplementary material, make clear titles for each part and connect them with the text. If the questionnaires are presented in the material, please translated them to English and write a caption on each. Make sure to mark the questions in them the same way they are in the main text, in the results etc.

Please look at the highlighted areas in the text and look for my comments, carefully.

Author Response

We thank the reviewer for his really valuable comments. We have modified the text trying to punctually follow the reviewer's suggestions. Thanks to the suggested changes, we believe that the text has improved significantly, and we hope that this revised version of the manuscript will meet the approval of the reviewer, the editor and therefore the readers. Thank you.

Reviewer 3 Report

Dear Authors,

The manuscript (ijerph-1210048) presented for review is very interesting, as well as a nutrition intervention conducted in the study. I recommend the article for publication after minor revision.

The study was finished in 2017 year, why Authors had waited to publication these results?

Authors, Please note and address the following comments:

Materials and Methods

In this section is the lack details of information about lessons for children. In supplementary material, I didn’t find this information.

Results

Lines 178-196: The way of counted points in Questionnaire A isn't clear for me.

Table 2 isn't understood for me. Below Table 2 should be informed about maximum points.   This was not reported anywhere in the manuscript.  It is important to add too in Table or below Table what is T0, T1, T2, T3 (readers have to look for this information in section Material and methods). In my opinion, the Title of this Table is shouldn't be: "Questionnaire B. Mean score ....." (Lines 217-218).  Title "mean score", should move to the content of the Table, and Authors should change the title of Table. I reckon the title of Table 2 will be better when we use for example Effect of nutrition education on food preferences by children (if I good understand the results presented in Table 2).

Discussion

This section is very modest. In my opinion, authors should use more articles to Discussion their results.

References

Authors, Please check the correctness of the citation of references by requirements of the International Journal of Environmental Research and Public Health. In my opinion, now it is not correct. The name of the journal is not italicized, there isn't doi. number.

Are the all names of authors are given in reference 11?

In the references, two of the literature source (positions 8 and 24), in my opinion, should be changed to a newer one.

Despite my comments, I am pleased to recommend this manuscript for publication. I believe it addresses an important area of research in an international context.

Best of luck with your paper and be safe!

Reviewer

Round 2

Reviewer 1 Report

Comments:  The article is much improved and I appreciate the authors’ attention to my comments in the revision and their feedback.  A few questions/comments remained unanswered that I believe need to be addressed. 

Major Concerns:

Lines 57 – 59 – here would be a good place to define “early” with some justification that then provides the rationale for the intervention’s focus on 8-9 year olds.  How does this fit with literature about when children’s food preferences are established?  Maybe lack of intervention impact is because the intervention needs to occur much earlier (preschool years).

Lines 122 – 144 – Some evidence in the form of a citation is needed to support the claim of “validated” for the questionnaires. If there is no peer-reviewed source, or any source, much more information needs to be given to support the validation claim.  How were they validated?  What type of validation (face, construct, etc), what procedures and samples were used for validation?  Were the factors identified by the authors the same as the ones in the validation study?  To me, it seems that the current study is in fact a validation study in addition to an assessment of the intervention.

Lines 441 -446  Your data still do not support this assumption.  Either reword – eg. One possibility for these poor results is ….” or delete.  I would also encourage that authors to discuss the possibility that the may intervention occur too late in the development of child food preferences and that may explain its lack of effects.

At the end of the day, the intervention produced one short-term effect on food knowledge and no effect on food attitudes, eating habits, or recognition of unhealthy foods.  While I appreciate the desire to provide ideas about ways to make it more effective or longer lasting, some discussion of the lack of effect and what that means is needed.  For example, do the study results support continued use of Edueat?  Might these findings, in conjunction with the other cited studies  about lack of impact, mean that school nutrition projects are not an effective approach (regardless of whether parents are involved or not)?

Minor concerns:

Overall, editing is needed to address some sentence errors.

Page 5 Lines 208 -211 – This seems to be an incomplete sentence

Page 8 – Lines 311 – “is” should be “are”

Lines 373-379 – Your lack of results are around eating habits would appear to differ from studies that find an increase in fruit consumption so a discussion of why food preferences changed but eating habits did not would be relevant here.

Lines 386-388 – This sentence needs clarification – maybe change to “increased knowledge of very healthy foods” are in line with…

Author Response

We thank the reviewer and are pleased that he appreciated our effort to edit the paper. We reiterate our belief that all of the reviewer's valuable suggestions significantly improved the manuscript. Also in this phase we have tried to accept the reviewer's suggestions, always motivating our choices. Thank you

Reviewer 2 Report

Dear authors,

The manuscript has significant improvements.  

Please, refine the language style and grammar in entire text.

For instance: 

Line 325 participant -> participants;

Sentence in lines 386-388 needs rephrasing as it is not clear, and so on. 

The supplement is significantly improved.

All the best.

Author Response

We thank the reviewer and are pleased that he appreciated our effort to edit the paper. We reiterate our belief that all of the reviewer's valuable suggestions significantly improved the manuscript. With this in mind, we have also accepted the suggestions in this second round.

Thank you.

Round 3

Reviewer 1 Report

The manuscript has been improved and I have only minor comments.

Line 57-60 – This sentence is confusing.

Lines 60-71 - The addition of this information helps to clarify your target age.  I think this would be clearer if you began a new paragraph with this sentence.  Please consider changing the discussion about "best" time to intervene to effective points to intervene (since we know that earlier then 8-9 is best but 8-9 might also be an effective time).  The rest of the paragraph needs rewriting  - avoid the use of "we", and provide a citation for the national and international guidelines you reference.

Methods section:  Instead of providing evidence of validation as a supplemental attachment, all the authors need to do is add a numbered citation in the body of the manuscript (the methods section) with corresponding information in the reference list.  Since there does not seem to be a peer-reviewed validation study, the authors need to reword the methods section to state that there is preliminary evidence of validation to avoid overstating the validity of these questionnaires.

I appreciate the changes you made to Lines 451-3.  I realize my comment needs further clarification. The sentence reads ‘These data suggest that one possibility for these poor results obtained from the…’

Does the term “these data” refer to the data provided by the study?  If that is the case, then “these data suggest that” should be deleted as your actual data do not investigate the impact of family involvement or school centeredness.  If “these data” refer to the studies cited in the preceding paragraph, then please consider rewording to “these studies” or “this research.”

Author Response

We thank the reviewer once again for allowing us to improve and refine our paper. Thank you!
